# Development of an Image De-Noising Method in Preparation for the Surface Water and Ocean Topography Satellite Mission

**Laura Gómez-Navarro** [1,2,*], **Emmanuel Cosme** [1], **Julien Le Sommer** [1], **Nicolas Papadakis** [3] and **Ananda Pascual** [2]

1   Institut des Géosciences de l'Environnement (IGE), Univ. Grenoble Alpes, CNRS, IRD, Grenoble INP, 38000 Grenoble, France; emmanuel.cosme@univ-grenoble-alpes.fr (E.C.); julien.lesommer@univ-grenoble-alpes.fr (J.L.S.)
2   Oceanography and Global Change, Institut Mediterrani d'Estudis Avançats (IMEDEA) (CSIC-UIB), 07190 Esporles, Spain; ananda.pascual@imedea.uib-csic.es
3   Institut de Mathématiques de Bordeaux (IMB), Univ. Bordeaux, Bordeaux INP, CNRS, UMR 5251, F-33400 Talence, France; nicolas.papadakis@math.u-bordeaux.fr
*   Correspondence: lauragomnav@gmail.com

**Abstract:** In the near future, the Surface Water Ocean Topography (SWOT) mission will provide images of altimetric data at kilometric resolution. This unprecedented 2-dimensional data structure will allow the estimation of geostrophy-related quantities that are essential for studying the ocean surface dynamics and for data assimilation uses. To estimate these quantities, i.e., to compute spatial derivatives of the Sea Surface Height (SSH) measurements, the uncorrelated, small-scale noise and errors expected to affect the SWOT data must be smoothed out while minimizing the loss of relevant, physical SSH information. This paper introduces a new technique for de-noising the future SWOT SSH images. The de-noising model is formulated as a regularized least-square problem with a Tikhonov regularization based on the first-, second-, and third-order derivatives of SSH. The method is implemented and compared to other, convolution-based filtering methods with boxcar and Gaussian kernels. This is performed using a large set of pseudo-SWOT data generated in the western Mediterranean Sea from a $1/60°$ simulation and the SWOT simulator. Based on root mean square error and spectral diagnostics, our de-noising method shows a better performance than the convolution-based methods. We find the optimal parametrization to be when only the second-order SSH derivative is penalized. This de-noising reduces the spatial scale resolved by SWOT by a factor of 2, and at 10 km wavelengths, the noise level is reduced by factors of $10^4$ and $10^3$ for summer and winter, respectively. This is encouraging for the processing of the future SWOT data.

**Keywords:** SWOT; de-noising; variational regularization; western mediterranean

## 1. Introduction

The Surface Water Ocean Topography (SWOT) [1] mission will provide an unprecedented two-dimensional view of ocean surface topography at a pixel resolution of 2 km. The launch is scheduled for 2021. SWOT's wide-swath altimeter, based upon Synthetic Aperture Radar (SAR) interferometry technology, will measure Sea Surface Height (SSH) over a 120-km wide swath with a 20-km gap at the nadir. The satellite will also carry a conventional nadir altimeter. SWOT will evolve on two different orbits: the first 3 months of scientific data production will be dedicated to a fast-sampling phase, where the repeat cycle will be 1 day. Then, the satellite will be moved to its

nominal orbit with a 20.86-day repeat cycle. SWOT is a multidisciplinary hydrology and oceanography mission, and here, we focus on the latter.

The main oceanographic objective of SWOT is to observe the geostrophic fine-scale circulation at the global scale [2,3]. The measurement system is designed to resolve ocean circulation patterns at scales down to 15 km, whereas the present-day constellation of conventional altimeters only resolves scales of 150–200 km and above [3]. In addition to potentially unexpected discoveries, this order-of-magnitude gain in resolution will help quantify several oceanic processes much more accurately than today. Among those processes are vertical motions, which are key to the vertical exchanges between the ocean surface and the atmosphere and between the ocean surface and the deep ocean [4–9], and the dissipation of kinetic energy, which partly determines the climatic role of the global ocean [10,11].

The SWOT mission objectives will be reached if we can accurately estimate gridded maps of at least the first- and second-order horizontal derivatives of SSH. Altimetry describes the upper ocean dynamics through geostrophy, which involves the horizontal SSH gradients. Geostrophy is a fairly good approximation of meso-scale dynamics, i.e., at scales larger than the first Rossby deformation radius (about 10–15 km in our region [12]), for which Rossby numbers are typically smaller than 1. Kinetic energy dissipation is driven by the horizontal strain rates of the ocean surface flow [13]. Complete, gridded maps of SSH derivatives are required for climate studies and short-term operational applications. One way to make gridded maps from incomplete SSH observations (including SWOT, but not only) is to assimilate those data into dynamical models. The assimilation of SWOT is expected to be challenging because of the spatially correlated errors, and promising solutions to this rely upon the joint assimilation of SSH and its derivatives [14,15]. All these considerations compel the scientific community to strive for getting accurate estimates of SSH derivatives.

Unfortunately, SWOT data will very likely be contaminated by small-scale noise and errors that prevent the direct computation of SSH derivatives. The errors expected to contaminate SWOT measurements gather several components with different spatial coherences and different amplitudes. Details are provided in the SWOT mission performance and error budget document [16]. To be prepared to exploit the future SWOT data, the SWOT simulator for ocean science has been developed to simulate realistic realizations of SWOT uncertainties [17]. Some are illustrated at our study region (Figure 1) in Figure 2. Errors due to the satellite roll, the baseline dilation, and the path delay induced by atmospheric humidity exhibit significant spatial correlations with different characteristic patterns. The system timing error presents errors invariant across-track but with possible small-scale variations along-track. The KaRIn (Ka-band Radar Interferometer) noise is spatially uncorrelated, with higher amplitudes at nadir and near the edges of the swath. The path-delay component also exhibits small-scale variations due to sharp changes in air humidity. Efforts have already been undertaken to filter out SWOT's random, small-scale noise by Gómez-Navarro et al. [18]. The authors show that the implementation of a diffusion-based filter allows to retrieve the dynamical spectral signature down to 40–60 km scales (20–30 km in terms of dynamical pattern scales). However, the de-noising approach here is not specifically designed to retrieve SSH derivatives, and we believe there is room for improvement in the scales to be retrieved.

This paper presents a method designed to remove the random, small-scale noise of the future SWOT data, which explicitly relies upon the regularity (bounded variations) of the first three orders of SSH derivatives. Consequently, this approach is of interest as it has a direct impact on not only SSH but also on important oceanic variables like geostrophic velocity and vorticity. This de-noising method is rooted in image restoration techniques of the variational type [19–22]. The range of image restoration techniques is extremely wide and diversified. Testing all existing methods is out of reach and irrelevant here. Our approach is then to acknowledge that our image is a smooth physical field with relatively smooth derivatives and that the estimation of derivatives is an important issue. This consideration guides the design of the de-noising method presented in Section 2. The method involves a set of parameters that must be adjusted. An essential task is to identify optimal sets of parameters. This study suggests a methodology to identify them. The experimental setup is described in Section 3. Sections 4

and 5 present the results, and Section 6 summarizes the study, draws the most relevant conclusions, discusses them, and suggests possible future research paths.

## 2. Variational De-Noising of SWOT Images with Penalization of Derivatives

### 2.1. Formulation of the Image De-Noising Problem

The purpose of image de-noising here is to allow the computation of first- and second-order SSH spatial derivatives of SWOT data as accurately as possible. The two reasons already mentioned in the introduction are (i) that these quantities represent geostrophic velocities and relative vorticity, respectively, of which the estimation is central to the success of SWOT mission, and (ii) that these quantities could be used to draw maximum benefits from the assimilation of SWOT data into ocean circulation models [14,15]. We therefore propose a method that explicitly constrains these derivatives.

The proposed de-noising model is formulated as a regularized least-square problem with a Tikhonov regularization. The de-noised SWOT image $h$ is searched for by minimizing the following cost function:

$$J(h) = \frac{1}{2}\|m \circ (h - h_{obs})\|^2 + \frac{\lambda_1}{2}\|\nabla h\|^2 + \frac{\lambda_2}{2}\|\Delta h\|^2 + \frac{\lambda_3}{2}\|\nabla \Delta h\|^2 \tag{1}$$

where $\|\quad\|$ represents the $L_2$-norm, $h^{obs}$ is the original noisy image (i.e., our observation, the pseudo-SWOT data), $\nabla = (\partial/\partial x, \partial/\partial y)$ is the gradient operator, and $\Delta = \partial^2/\partial x^2 + \partial^2/\partial y^2$ is the Laplacian operator. Letter $m$ and sign $\circ$ represent a mask and the entrywise matrix product, respectively. They can be ignored for the present and the next subsection: their role is discussed in Section 2.3 below. The regularization terms impose regularity constraints on geostrophic velocity, vorticity, and vorticity gradient, respectively. Parameters $\lambda_1$, $\lambda_2$, and $\lambda_3$ must be prescribed. The search for their optimal values is reported in Section 3.3.

### 2.2. Resolution of the Variational Problem

The variational problem displayed in Equation (1) is solved using a gradient descent method [23]. The gradient of $J$ is written as follows:

$$\nabla J(h) = m \circ (h - h_{obs}) - \lambda_1 \Delta h + \lambda_2 \Delta \Delta h - \lambda_3 \Delta \Delta \Delta h \tag{2}$$

so that the solution can be reached after convergence of the following iterations:

$$h^{k+1} = h^k + \tau \left( m \circ (h_{obs} - h^k) + \lambda_1 \Delta h^k - \lambda_2 \Delta \Delta h^k + \lambda_3 \Delta \Delta \Delta h^k \right) \tag{3}$$

Stability of iterations is guaranteed if $\tau \leq (1 + 8\lambda_1 + 64\lambda_2 + 512\lambda_3)^{-1}$. In practice, it is taken equal to this value. Two improvements on the method's implementation accelerate the gradient descent: Firstly, iterations are started with a preconditioned image obtained by applying a Gaussian filter onto the original image, including inpainting as discussed in Sections 2.3 and 2.4 (note that $h^{obs}$ remains the original, unfiltered image). Preconditioning considerably speeds up the algorithm convergence, in particular for the inpainted regions. Secondly, iterations are actually implemented with an acceleration of Scheme 3 based on the Fast Iterative Shrinkage-Thresholding Algorithm (FISTA) [24], detailed in Appendix B. Iterations are stopped when $\|h^{k+1} - h^k\| < 10^{-9}$ or if $k = 10^4$. Those values have been fixed after a careful search of a trade-off between accuracy and numerical efficiency.

The Laplacian operator is discretized with finite differences using the five-point stencils of the image pixels. As commonly done in image processing, the division by pixel size is ignored; this also reduces the probability of truncation errors due to operations with terms different by too many orders of magnitude. Pixels located at the boundaries, where the stencil is incomplete, must have a Laplacian value attributed; otherwise, the image would become smaller at each iteration of the gradient descent. The implementation of the Laplacian operator follows Chambolle [22] and is detailed in Appendix A.

### 2.3. Dealing with Gaps in the Image

An inpainting method is implemented to deal with islands, continents, and the 20-km wide gap at the SWOT nadir, which all represent obstacles to the calculation of the second derivatives of images. Inpainting consists of filling the gaps consistently with the neighbouring water pixels. This is done (i) by extending images $h^{obs}$ and $h$ with pixels in the gaps and (ii) by filling mask $m$ with ones in the water pixels of the original image and zeros in the gaps. Differential operators can then be applied to every image pixel, and the gradient descent iterations are carried out smoothly. Mask $m$ is applied to the resulting image to obtain the final, filtered image with islands, continents, and the nadir gap.

Inpainting should not only be considered as a complimentary step to facilitate the gradient descent implementation but also as an opportunity to fill the nadir gap for calibration, validation, and reconstruction purposes. In the gaps, the image resulting from the iterations is determined only by the neighbouring water pixels and regularity constraints. The gap width (20 km) appears reasonably small in comparison with spatial scales of SSH variations in most parts of the mid-latitude, open ocean. The image values obtained at nadir may thus be comparable to those collected by the nadir instrument carried by SWOT, allowing calibration of the radar interferometer, validation of data, and reconstruction of SSH in gap-free images. Such opportunities will be explored in a future work.

### 2.4. Comparison with Convolution-Based Filters

In Section 4, the image de-noising technique described above will be compared with standard-type filters, namely convolution-based filters. In our experiments, we test the two commonly used boxcar and Gaussian convolution kernels, with a large range of parameters, and we shortly refer to the boxcar filter and the Gaussian filter. Their parameters are the box size (or footprint) and the standard deviation for the Gaussian kernel (hereinafter referred to as $\sigma$). Gaps in the SWOT swath (lands, islands, and nadir gap) are inpainted to facilitate filtering and to ensure the smoothness of SSH fields. Then, SSH values created in gaps are removed for the evaluation of the methods using the mask $m$. Inpainting is implemented as follows: (i) Image gaps are filled with zeros; (ii) both the filled image and the mask $m$ are filtered with the same kernel; and (iii) the filtered filled image is divided entrywise by the filtered mask. Note that, in an earlier study [18], a Laplacian diffusion filter was experimented. It is not reproduced here, since it is equivalent to the Gaussian filter implemented in this study. For details on the software used to implement these methods see Appendix E.

## 3. Experimental Setup

### 3.1. Simulated SWOT Dataset

The input of our database is a 15-month North Atlantic simulation at a resolution of $1/60°$. We use the NEMO3.6 ocean model coupled to LIM2 ice model, with atmospheric forcing from a global ocean reanalysis at $1/4°$ (GLORYS-v3) and ocean-atmosphere boundary conditions of Drakkar Forcing Set (DFS5.2) based on European Centre for Medium-Range Weather Forecasts (ECMWF) Interim Re-Analysis (ERA-Interim). It has no high frequency forcing and thus does not include tides. The domain covers the North Atlantic from 25°N to 66°N. The horizontal resolution is between 0.8 and 1.6 km (depending on latitude), and the grid has 300 vertical levels. This NEMO (Nucleus for a European Model of the Ocean) model configuration is referred to as NATL60, and the source files and codes are available in Molines [25]. The particular simulation used herein has been described in Amores et al. [26], Fresnay et al. [27], Ajayi et al. [28]. Lastly, the simulation time span is from mid-June 2012 to October 2013 [29].

The SWOT simulator for Ocean Science (version 2.21) [17] is run to generate pseudo-SWOT scenes from the NATL60 simulation. The SWOT simulator first builds the SWOT observation grid based on the provided satellite orbit. In this study, SWOT grid resolution is fixed at 1 km. After this work started, the resolution of the basic SWOT level 2 SSH data products has been fixed to 2 km, but this

small mismatch does not modify the general approach. After building the grid, the simulator reads SSH data from NATL60 and linearly interpolates them from the model to the SWOT grid (rendering the variable SSH_model). In a last step, it computes random realizations of observation errors and adds them to the interpolated SSH data (rendering SSH_obs). Observation errors considered at the moment are KaRIn noise errors, roll errors, phase errors, baseline dilation errors, timing errors, and errors due to signal alternation by atmospheric humidity. Among these errors, only the KaRIn noise is expected to be spatially uncorrelated. Technically, the SWOT simulator provides simulations of the noise-free SSH observed by SWOT and of the noisy data that SWOT will actually yield (sum of the former and the noise: SSH_obs = SSH_model + errors). For the evaluation of image de-noising methods, it thus provides "true" noise-free images ($h^{true}$) along with the realistic SWOT data ($h^{obs}$) to process and compare with the truth.

A set of 543, $121 \times 200$ km$^2$ pseudo-SWOT scenes are generated in the western Mediterranean Sea, covering one winter and two summer seasons (choice limited by the model's time span). SWOT scenes are sampled from the fast-sampling phase satellite orbit, focusing on a cross-over region, i.e., where an ascending pass crosses a descending pass, therefore providing 2 passes per day. The SWOT data simulation is carried out over three 3-month periods: July to September 2012 and 2013 (JAS12 and JAS13 hereafter), representing the summer season, and February to April 2013 (FMA13), representing the winter season. The summer periods provide 92 (resp. 91) of ascending (resp. descending) passes; the winter period provides 89 (resp. 88) passes. The selected region belongs to the fast-sampling phase crossover in the western Mediterranean Sea. This is one of the regions selected for calibration/validation (Cal/Val) [30] in which *in situ* measurements have been made in the frame of SWOT [31]. To mitigate the computational complexity of the study and to avoid the presence of continents and islands, limited subregions of the SWOT swaths are sampled. These subregions are 121-km wide (the width of 2 SWOT swaths plus the gap) and 200-km long. The region, the SWOT passes, and the subregions are shown in Figure 1. It is worth noting that each scene is affected by a unique realization of the SWOT error.

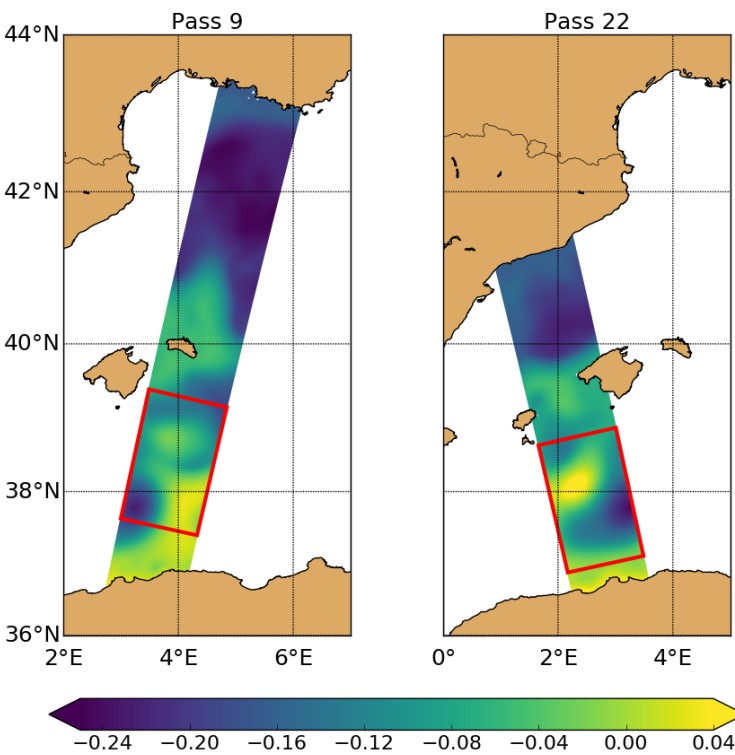

**Figure 1.** SSH_model outputs (m) for cycle 1 of pass 9 (**left**) and 22 (**right**) of the July to September 2012 (JAS12) dataset: In red is the subregions selected.

In this work, image de-noising techniques are first applied to the pseudo-SWOT scenes affected by the KaRIn noise only (SSH_obs_K) and then to the scenes containing all errors (SSH_obs). This approach allows to discriminate the effects and the performance of image de-noising in the presence of the spatially correlated SWOT errors. A few realizations of the different components of the SWOT error are shown in Figure 2, where we can observe how most errors exhibit strong and long-range correlations, whilst the KaRIn error does not show any correlation at all.

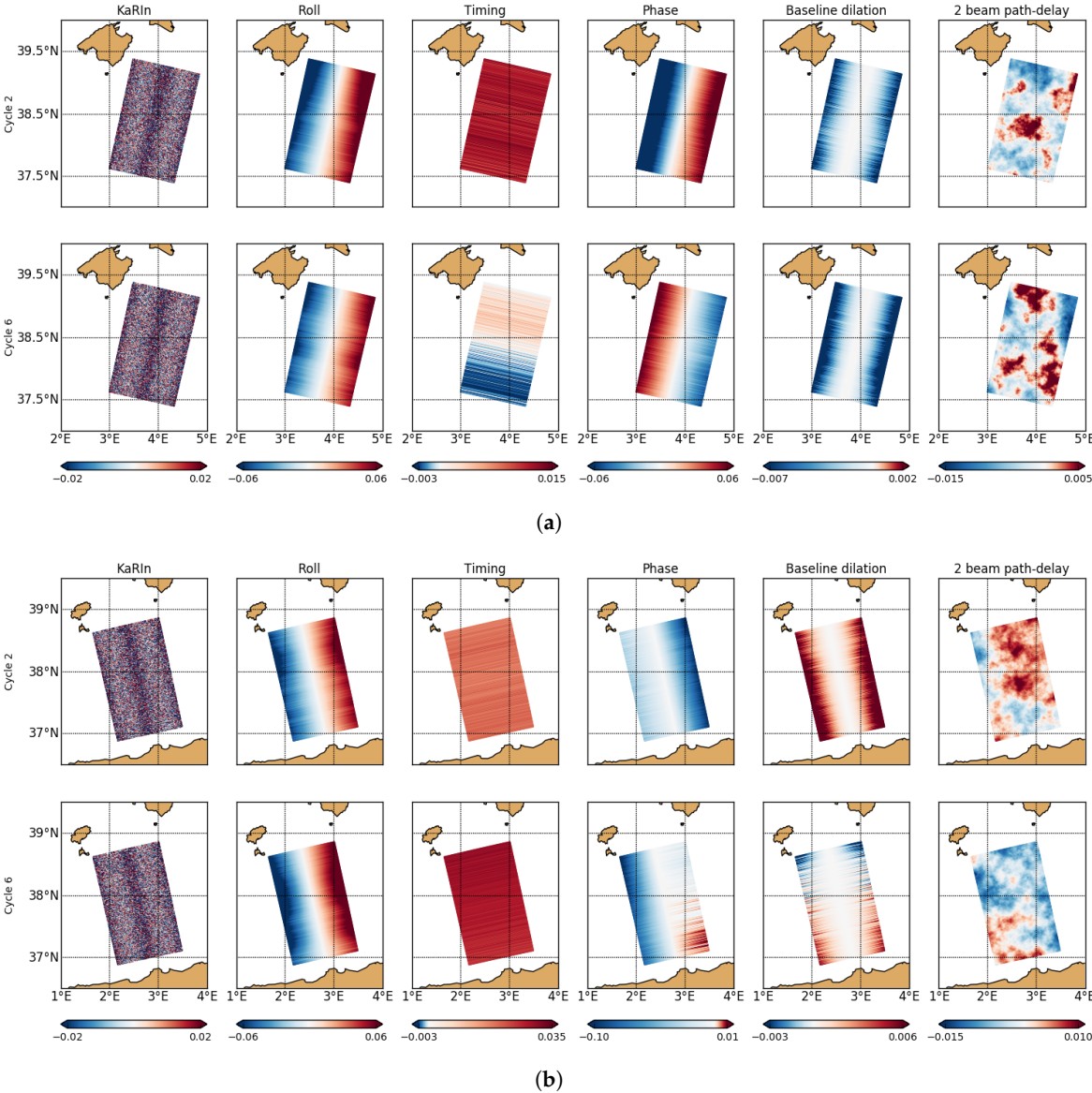

**Figure 2.** Examples of errors (m) added by this Surface Water Ocean Topography (SWOT) simulator version 2.21 to our study region fast-sampling phase for JAS12 pass 9 (**a**) and 22 (**b**): Note that these simulations are performed without the 20-km gap at nadir.

### 3.2. Diagnostics for Evaluation

The quantitative evaluation of de-noising methods is carried out by computing Root Mean Square Errors (RMSE) and Mean Spectral Ratios (MSR). RMSE for a single de-noised SWOT field $h$ is computed as the Euclidean distance to the corresponding original, noise-free field $h^{\text{true}}$:

$$RMSE(h) = \sqrt{\frac{1}{N} \sum_{i=1}^{N} \left(h_i - h_i^{\text{true}}\right)^2} \tag{4}$$

where $N$ is the number of pixels and $i$ a pixel index. Single image RMSEs are then averaged out by season for the 3 seasons considered and are computed for SSH, $|\nabla \text{SSH}|$ and $\Delta \text{SSH}$. Thus, the test of a de-noising method with a specific set of parameters results in 9 RMSE values. To evaluate the improvement after the application of the different de-noising techniques and parameters, we also calculate the percentage of the initial RMSE left. We calculate this RMSE residual ($RMSE_r$) as follows:

$$RMSE_r(h) = \frac{RMSE(h)}{RMSE(h^{obs})} \times 100, \tag{5}$$

where h is the de-noised field and $h^{obs}$ is the original noisy field (SSH_obs_K or SSH_obs).

The spatial spectra of the de-noised SWOT SSH are compared with the spectra of the noise-free and the noisy SWOT SSH. For each pass, we calculate the cross-track averaged and along-track power spectrum. The spectra are then averaged out over each season, leading to one spectrum per season. Information on the wavenumber spectrum calculations is given in Appendix C. Again, to evaluate the improvement after the application of the different de-noising techniques and parameters, we compare the noise-free and de-noised fields. To do so, the Mean Spectral Ratio (MSR) is computed from the Power Spectral Densities (PSD) of SSH. For each season, MSR is computed as follows:

$$MSR = \sqrt{\frac{1}{\sum_{j=1}^{N_k} \delta k_j} \sum_{j=1}^{N_k} \left( \left( \log_{10} \left( \frac{PSD_j(h^{\text{true}})}{PSD_j(h)} \right) \right)^2 \times \delta k_j \right)}, \tag{6}$$

where $N_k$ is the number of wavelengths considered and where $PSD_j(h^{\text{true}})$ and $PSD_j(h)$ are the power spectral density values at wavelength $j$ for the original, noise-free SWOT field and the de-noised SWOT field, respectively. The considered wavelengths span the interval from 9 km, the approximate effective resolution of NATL60, to 200 km, the size of images along-track. MSR is defined above so that the best score is 0.

*3.3. Exploring Parameters of the De-Noising Methods*

For all de-noising methods, a wide range of parameters are tested to identify optimal parameters according to the diagnostics presented in Section 3.2. The convolution-based methods use a single parameter that can easily be compared with the image dimensions in pixels. For the boxcar kernel, the tested parameter values go from 3 to 200 km and correspond to the size of the box in pixels (1 km in our case). For the Gaussian kernel, the tested parameters go from 0.25 to 300 and correspond to the standard deviation, in pixels (we test up to a big sigma to have a highly oversmoothed image to reach the limit of the method). On the contrary, the geometric interpretation of the parameters of the variational method is not straightforward, and a wide exploration of the parameter space must be undertaken. However, due to computation time limitations, this cannot be performed in a strictly systematic manner. The adopted procedure is detailed below.

3.3.1. Orders of Magnitude of the Cost Function Terms

The orders of magnitude of the terms $\|\nabla h\|^2$, $\|\Delta h\|^2$, and $\|\nabla \Delta h\|^2$ composing the cost function (Equation (1)) are estimated to coarsely scale the parameters $\lambda_1$, $\lambda_2$, and $\lambda_3$. The rationale is that, for one of these terms (with its weight) to have some impact on the solution, it must be of an order of magnitude not too different from the background term $\|m \circ (h - h^{obs})\|^2$. Figure 3 shows the seasonal evolution of the derivative terms, computed from the model in a $2° \times 2°$ region containing the SWOT passes used in this study. The relative ratios between $\|\nabla h\|^2$, $\|\Delta h\|^2$, and $\|\nabla \Delta h\|^2$ are approximately 1000:10:1. Therefore, if we want to include all three terms in the cost function, the ratios between $\lambda_1$,

$\lambda_2$, and $\lambda_3$ should coarsely be 1:100:1000. Those ratios must be only considered as a guideline to start the investigation, not a strict rule. Note that the order of magnitude of the background term after minimization of the cost function is thought to be in the range 1 to 100 in the same region. This has been estimated using the noise-free field.

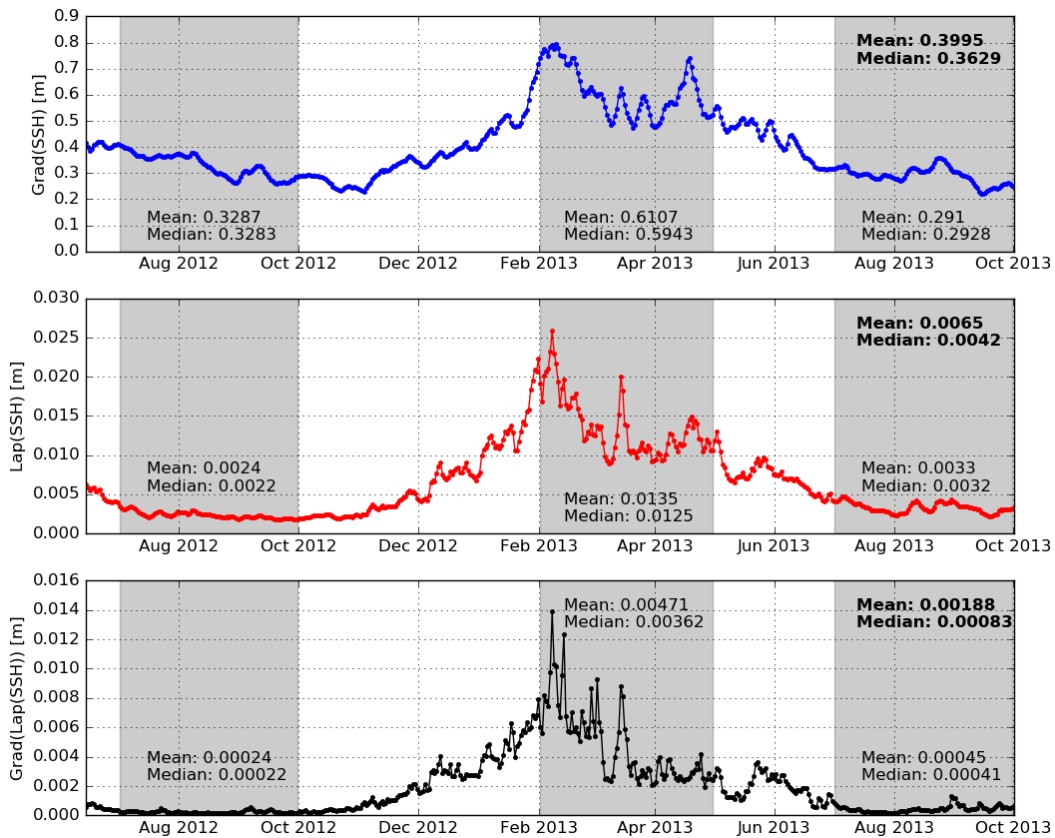

**Figure 3.** Seasonal variations of the cost function terms $\|\nabla h\|^2$, $\|\Delta h\|^2$, and $\|\nabla \Delta h\|^2$, from top to bottom: Shaded areas indicate the JAS12, February to April 2013 (FMA13), and July to September 2013 (JAS13) periods from left to right, respectively. The mean and median values are printed for each period and for the whole year (upper right corners, in bold).

### 3.3.2. Finding Optimal Sets of Parameters

First, we created an exponential series of values to be tested for the three lambdas, consistently with the previously estimated relative ratios. For $\lambda_1$, the series is chosen as $\{4^n, n = 0, ..., 7\}$. For $\lambda_2$ and $\lambda_3$, the series are $\{10 \times 4^n, n = 0, ..., 7\}$ and $\{100 \times 4^n, n = 0, ..., 7\}$, respectively. With these, six scenarios of Derivatives Penalization (DP) de-noising are investigated, including one, two, or three penalization terms in the cost function of Equation (1). Three scenarios out of the six considered include a single penalization term (mono-parametric) of orders 1, 2 and 3, successively. The other scenarios are made of terms of orders 1 and 2 and of 2 and 3, and the last one includes the three orders. For conciseness, particularly in the next section, we refer to the variational method with the first-order term only as the $\lambda_1$ method. We similarly refer to the $\lambda_2$ method and to the $(\lambda_1 + \lambda_2)$ method when the first two penalization terms are considered, and so on.

For each scenario, a two-step procedure is implemented to identify an optimal set of parameters. In a first step, de-noising of the full set of images is performed with all possible combinations of parameters permitted by the scenario and the parameter series defined previously. RMSEs and MSR are computed for all the combinations. In a second step, refined series of parameters are created in

the neighborhood of the combination of parameters that yields the minimum RMSE and MSR scores. Image de-noising is then carried out again with all possible combinations of these series.

## 4. Optimal De-Noising Method

In this section, the optimal de-noising method is searched for based on the RMSE and MSR scores described in Section 3. We investigate the KaRIn-noise-only scenario and then the all errors scenario and finally have a closer look at the method identified as optimal. As it becomes clear in what follows, the notion of optimality does not only refer to quantitative measures. The design of a single index summarizing the performance of the method for the different RMSEs is indeed subjective. Moreover, we take into account the ease of implementation and parameterization as a criteria in the final decision.

Minimum values of RSME and MSR for each season, method, and variable are reported in Tables 1 and 2 for the KaRIn-only and all errors scenarios, respectively. RMSE scores are actually expressed as the percentage of the original RMSEs ($RMSE_r$), i.e., those of the original, noisy data. For SSH, $|\nabla SSH|$, and $\Delta SSH$ RMSE and MSR of each de-noising configuration and parameterization, the scores do not necessarily correspond to the same optimal parameter (box size, $\sigma$, or $\lambda$).

**Table 1.** Scores summary of boxcar, Gaussian and Derivatives Penalization (DP) de-noising methods for the just KaRIn (Ka-band Radar Interferometer) dataset.

| Season | De-noising method | | RMSE$_r$ | | | Minimum MSR |
|---|---|---|---|---|---|---|
| | | | SSH | $|\nabla$SSH$|$ | $\Delta$SSH | |
| JAS12 | | Boxcar | 12.43 | 0.094 | 0.300 | 0.2010 |
| | | Gaussian | 11.23 | 0.067 | 0.250 | 0.1111 |
| | DP | 1 | 12.55 | 0.084 | 0.279 | 0.2028 |
| | | 2 | 08.71 | 0.050 | 0.247 | 0.0143 |
| | | 3 | 09.06 | 0.051 | 0.247 | 0.1021 |
| | | 1 + 2 | 08.72 | 0.050 | 0.247 | 0.0192 |
| | | 2 + 3 | 08.68 | 0.049 | 0.247 | 0.0205 |
| | | 1 + 2 + 3 | 08.66 | 0.049 | 0.246 | 0.0259 |
| FMA13 | | Boxcar | 15.04 | 0.177 | 0.511 | 0.1066 |
| | | Gaussian | 12.97 | 0.133 | 0.424 | 0.0746 |
| | DP | 1 | 15.41 | 0.173 | 0.483 | 0.1498 |
| | | 2 | 10.92 | 0.115 | 0.420 | 0.0178 |
| | | 3 | 10.86 | 0.113 | 0.416 | 0.0682 |
| | | 1 + 2 | 10.92 | 0.115 | 0.420 | 0.0208 |
| | | 2 + 3 | 10.79 | 0.113 | 0.416 | 0.0168 |
| | | 1 + 2 + 3 | 10.82 | 0.113 | 0.416 | 0.0255 |
| JAS13 | | Boxcar | 11.98 | 0.086 | 0.326 | 0.1796 |
| | | Gaussian | 10.99 | 0.063 | 0.276 | 0.0911 |
| | DP | 1 | 12.78 | 0.083 | 0.309 | 0.2031 |
| | | 2 | 08.96 | 0.053 | 0.274 | 0.0216 |
| | | 3 | 09.11 | 0.053 | 0.273 | 0.1010 |
| | | 1 + 2 | 08.97 | 0.053 | 0.274 | 0.0394 |
| | | 2 + 3 | 08.84 | 0.052 | 0.272 | 0.0243 |
| | | 1 + 2 + 3 | 08.84 | 0.052 | 0.272 | 0.0269 |

### 4.1. RMSE and MSR Scores with KaRIn Noise Only

For all variables ($h$, $\nabla h$, and $\Delta h$), all seasons, and all de-noising methods, minimum RMSEs are smaller in summer than in winter (Table 1). This is expected because the oceanic surface features in

winter are smaller than in summer [32], so their observation is more affected by the KaRIn noise. Also, smaller structures are more affected by the smoothing due to the de-noising.

For all three seasons and all three variables, RMSEs and MSRs from the convolution-based methods and from the $\lambda_1$ method are larger than RMSEs and MSRs from all other variational methods. Also, the $\lambda_3$ method provides MSRs significantly higher than the other variational methods. None of these methods is the optimal de-noising one in this KaRIn-only noise configuration and are not further discussed in the following.

In terms of both RMSEs and MSRs and among the methods still on course, no method outperforms the others systematically and distinctly. For all three variables, RMSEs are close to each other, with differences less than a very few percents. MSRs are a bit more scattered but without any clear predominance of a specific method in all seasons. However, the $\lambda_2$ method exhibits the lowest MSR values in summer and the second lowest value in winter, close to the $\lambda_2 + \lambda_3$ method.

Finally, this analysis persuades us to further examine the $\lambda_2$ method for the KaRIn-only scenario (see Section 4.3). This choice is supported by the RMSE and MSR analysis above, which shows that other methods do not beat it clearly, and by the fact that it is much easier to parametrize a single-parameter method rather than a two- or three-parameter method.

### 4.2. RMSE and MSR Scores with All Errors

Normalized minimum RMSEs for $h$ and $|\nabla h|$ are higher than in the KaRIn-only scenario by factors of 6–12 for $h$ and 1.5–4 for $|\nabla h|$ (Table 2). This is obviously due to the spatially correlated component of the errors (see Figure 2), which is not filtered out by any of the methods used here. Other approaches must be used to remove the correlated errors in order to obtain more accurate estimates [33,34].

Contrary to $h$ and $\nabla h$, RMSEs for $\Delta h$ are comparable with those obtained in the KaRIn-only case. They are 5% higher only. This slight increase in RMSE is the signature of the nonlinear component of the correlated error. In the across-track direction, this nonlinear (quadratic, more precisely) component is due to the baseline dilation [16,17]. The other components are constant, linear, or piecewise linear and thus are removed by the second-order derivatives. The spatial errors' signal in the along-track direction is supposed to vary less with the satellite attitude (except timing errors) and the local environmental constraints on the satellite (baseline dilation). This is reflected as small but not null second-order derivatives of errors that may slightly affect the RMSE. In reality, the spatial decorrelation of such errors might be short and combined with geophysical variations (waves and atmosphere).

Considering only RMSEs on $\Delta h$, except for the boxcar and the $\lambda_1$, no method performs significantly better than the others, and RMSEs are higher in winter than in summer. This is similar to the KaRIn-only scenario. The Gaussian filter performs comparatively better than in the KaRIn-only scenario.

In terms of MSRs, the methods involving $\lambda_2$ perform significantly better than the others, including the $\lambda_3$ and the Gaussian methods. These last two exhibit MSRs larger than the others by factors of 1.5 to 4. In winter, the $\lambda_2$ method is a little less effective than the multi-parameter methods, with a MSR twice as large.

The de-noising experiments with all errors, like those with the KaRIn noise only, lead us to favor the $\lambda_2$ method. The reasons are similar: based on RMSEs and MSRs, the method compares favorably with the others, and a single-parameter method is much easier to parametrize. The only result speaking against this choice is the MSR in winter, for which the multi-parameter methods perform better than the $\lambda_2$ method. Considering the score value though and after the examination of the wavenumber spectra (see Section 5) this point hardly justifies discarding the $\lambda_2$ method as the optimal one.

### 4.3. A Focus on the Second-Order Variational Method

This section investigates the sensitivity of the $\lambda_2$ de-noising to the parameter value. Figure 4 shows the RMSEs for $h$, $|\nabla h|$, and $\Delta h$ and the MSR for $h$ as functions of $\lambda_2$. On each graph, the three seasons are shown for both KaRIn-only (solid lines) and all errors (dashed lines) scenarios, making a total of 6 curves.

Except for $h$ and $|\nabla h|$ RMSEs in the all errors scenario, all RMSE and MSR curves exhibit a clear minimum point, which indicates the existence of an optimal or a range of close-to-optimal $\lambda_2$ values for the de-noising. These optimal values are larger in summer than in winter. This is very likely because small-scale dynamics are amplified in winter [10]. Large $\lambda_2$ values tend to over-smooth the SSH field in winter, leading to higher residual errors. The seasonal difference in optimal $\lambda_2$ values is particularly evident with MSR, with values of ~100 in winter and of ~400 in summer. In Figure 4, the impact of applying the optimal $\lambda_2$ value for winter to summer (or viceversa) can be observed. RMSEs are not very sensitive to $\lambda_2$ near the optimal values, contrary to MSRs. For instance, using the winter value in summer (or vice versa) barely changes RMSEs but affects MSR more significantly. RMSEs for $h$ and $\nabla h$ in the all errors scenario are dominated by the correlated SWOT errors, which remain present after de-noising. Consistently with the analysis of the previous section, those RMSEs are much higher in the all errors than in the KaRIn-only scenario.

**Table 2.** Scores summary of boxcar, Gaussian and Derivatives Penalization (DP) de-noising methods for the all errors dataset.

| Season | De-noising method | | RMSE$_r$ | | | Minimum MSR |
|---|---|---|---|---|---|---|
| | | | SSH | \|∇SSH\| | ΔSSH | |
| JAS12 | Boxcar | | 90.31 | 0.171 | 0.303 | 0.2024 |
| | Gaussian | | 90.10 | 0.156 | 0.264 | 0.1181 |
| | DP | 1 | 87.60 | 0.159 | 0.281 | 0.1922 |
| | | 2 | 90.57 | 0.174 | 0.261 | 0.0307 |
| | | 3 | 90.22 | 0.156 | 0.265 | 0.1359 |
| | | 1 + 2 | 87.61 | 0.158 | 0.262 | 0.0328 |
| | | 2 + 3 | 90.22 | 0.156 | 0.261 | 0.0391 |
| | | 1 + 2 + 3 | 87.40 | 0.156 | 0.261 | 0.0395 |
| FMA13 | Boxcar | | 90.88 | 0.250 | 0.511 | 0.1274 |
| | Gaussian | | 90.76 | 0.221 | 0.435 | 0.0515 |
| | DP | 1 | 89.89 | 0.237 | 0.484 | 0.1415 |
| | | 2 | 91.11 | 0.226 | 0.432 | 0.0314 |
| | | 3 | 90.96 | 0.226 | 0.432 | 0.0868 |
| | | 1 + 2 | 89.90 | 0.223 | 0.435 | 0.0160 |
| | | 2 + 3 | 91.00 | 0.226 | 0.430 | 0.0177 |
| | | 1 + 2 + 3 | 89.82 | 0.220 | 0.430 | 0.0203 |
| JAS13 | Boxcar | | 89.73 | 0.137 | 0.328 | 0.1792 |
| | Gaussian | | 89.18 | 0.126 | 0.289 | 0.1152 |
| | DP | 1 | 84.30 | 0.131 | 0.310 | 0.1895 |
| | | 2 | 90.36 | 0.142 | 0.287 | 0.0254 |
| | | 3 | 89.66 | 0.127 | 0.290 | 0.1251 |
| | | 1 + 2 | 84.19 | 0.131 | 0.287 | 0.0237 |
| | | 2 + 3 | 89.66 | 0.127 | 0.286 | 0.0285 |
| | | 1 + 2 + 3 | 83.75 | 0.127 | 0.286 | 0.0267 |

Finally, the optimal $\lambda$ can be defined within a range that is a compromise between the RMSE and MSR results, for each season. In Figure 4, horizontal bars indicate the range of $\lambda_2$ values that provide scores higher than the minimum by less than 5%. For each season, the overlap of all horizontal bars defines a range of optimal $\lambda_2$ values. Not detailed here, the results from the other (single or multiple-parameter) configurations of variational de-noising also exhibit such overlaps, except for $\lambda_1$. Based on this information and on MSR scores, we propose $\lambda_2$ intervals of [300–400] and [400–500] in summer for the KaRIn-only and all errors cases, respectively. With $\lambda_2$ in these intervals, MSR scores

remain close to their minima. Note that the two summer seasons' results render slightly different optimal values, suggesting that the optimal $\lambda_2$ choice is inevitably subject to a part of subjectivity if no additional information on the ocean surface dynamics is available. In winter, the optimal $\lambda_2$ interval is [90–120]. Due to more energetic dynamics that make the signal-to-noise ratio higher, the MSR optimal $\lambda_2$ values in the KaRIn and all errors scenarios are much closer to each other than in summer.

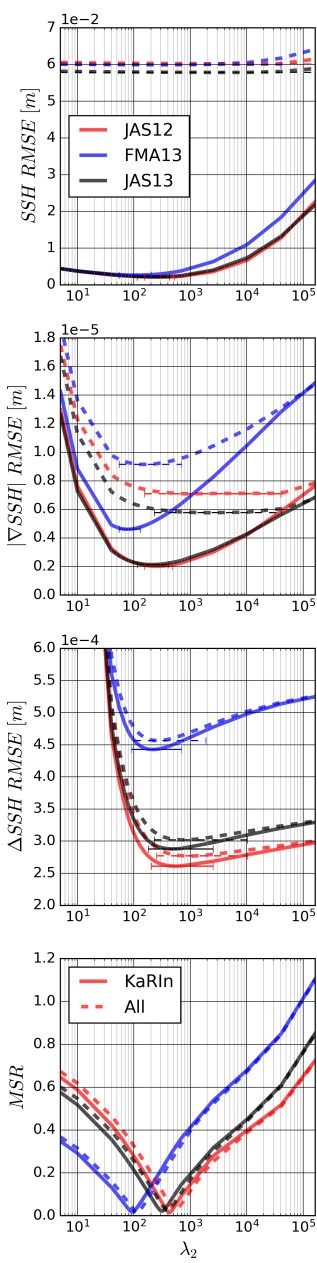

**Figure 4.** Scores of RMSE and MSR of the $\lambda_2$ method from just KaRIn (solid line) and all errors (dashed line) for all 3 seasons: Horizontal bars in the RMSE plots show the range of $\lambda_2$ values that provide scores higher than the minimum RMSE by less than 5%.

## 5. Retrieved SWOT Fields and Spatial Spectra

Figures 5 and 6 illustrate the de-noising with the $\lambda_2$ method on SWOT passes and are presented in the same format: $h$ on the top panel, $|\nabla h|$ on the central panel, and $\Delta h$ on the bottom panel. The first panel shows, from left to right, the original, noise-free $h$ field, $h$ with KaRIn noise only, $h$ with all sources of errors, the de-noised KaRIn-only $h$, and the de-noised all errors $h$. The de-noised data have

been obtained with the $\lambda_2$ method with parameter values chosen within the intervals identified in Section 4.3 and indicated on each graph. The second and third panels show the corresponding $|\nabla h|$ and $\Delta h$, respectively. Figure 5a,b exhibit summer time scenes with high and low correlated SWOT errors, respectively. Figure 6a,b are the corresponding winter plots.

In all cases, de-noising leads to correct orders of magnitude for all fields and particularly for $|\nabla h|$ and $\Delta h$. This is not the case for the conventional convolution-based methods (see Appendix D). As expected and already shown by Gómez-Navarro et al. [18] and Chelton et al. [35], the original SWOT data affected by random, small-scale noise does not provide any useful information about SSH derivatives. The de-noising method corrects this efficiently and makes it possible to identify the main structural characteristics of the fields.

A strong spatially correlated error shows strong signatures on $h$, moderate signatures on $|\nabla h|$, and low signatures on $\Delta h$, except at the outer boundaries of the swath. The low signature on $\Delta h$ was already observed in the RMSEs and is due to the specific spatial structure of the errors. Most components are linear in the across-track direction. In the along-track direction, the impact is lower because for the wavelengths impacted by the filtering, the error correlation is high (Figure 2). Therefore, the correlated errors have a low effect on the second-order derivatives. The remaining noise at the outer boundaries is due to the finite difference method used to compute the derivatives described in Appendix A.

Although the resulting fields of $\Delta h$ fall in correct orders of magnitude and capture the structure of the true fields at the scale of the swath, the kilometric-scale fronts and filaments are smoothed out by the de-noising. Solving this issue would require the development of more sophisticated de-noising techniques or a post-processing of the present result including, for example, some ocean dynamics through data assimilation techniques. This will be a natural step forward, since the first motivation for developing a de-noising technique constraining $\Delta h$ is precisely the combined assimilation of $h$ and its first two derivatives, as stated in the introduction.

Figure 7 shows Power Spectral Densities (PSD) of $h$. The rows distinguish the just KaRIn noise added and the all errors cases. The columns are for summer 2012, winter 2013, and summer 2013. On each graph, the spectra are shown for the noise-free data (SSH_model), the noisy data (SSH_obs), the de-noised data (SSH_obs_f), the pre-de-noising noise (noise), and post-de-noising noise (noise_f). The de-noised data have been obtained with the same $\lambda_2$ values as in Figures 5 and 6. Each spectra has an envelope showing PSD values between the 5th and 95th percentile, representing the PSD variability. This envelope reduces with the PSD values and with wavelength. At small scales, SSH_obs_f's envelope is narrower than SSH_model's, very likely because a fine-scale part of the physical signal is removed along with the noise.

From this spectral viewpoint, the de-noised data matches the noise-free data well at all scales down to ∼15 km. In the noisy data, the noise amplitude approaches the signal amplitude at wavelengths of 50 km in summer and 40 km in winter and dominates the signal at shorter wavelengths. This is efficiently corrected by the de-noising. The process seems more efficient in winter than in summer, probably because of higher PSDs in winter related to more intense ocean surface processes.

Following the definition proposed by Wang et al. [36] for the spatial scale resolved by SWOT, the de-noising reduces this scale by a factor of 2, leading to resolved scales between 20 and 30 km approximately. Wang et al. [36] define the spatial scale resolved by SWOT by the wavelength at which the SWOT noise spectrum intersects the spectrum of the true signal (SSH_model here). Figure 7 indicates resolved scales of 50, 40, and 50 km in the JAS12, FMA13, and JAS13 scenarios, respectively, in both just KaRIn and all errors cases. After de-noising, the resolved scales are reduced to approximately 25, 20, and 25 km in the KaRIn-only case and to 30, 20, and 30 km in the all errors case. Even below these scales, the noise left is very low and within the variability of SSH_model (red envelope in Figure 7). At wavelengths near 10 km, the noise is reduced by $10^4$ in the JAS scenarios and $10^3$ in the FMA scenario.

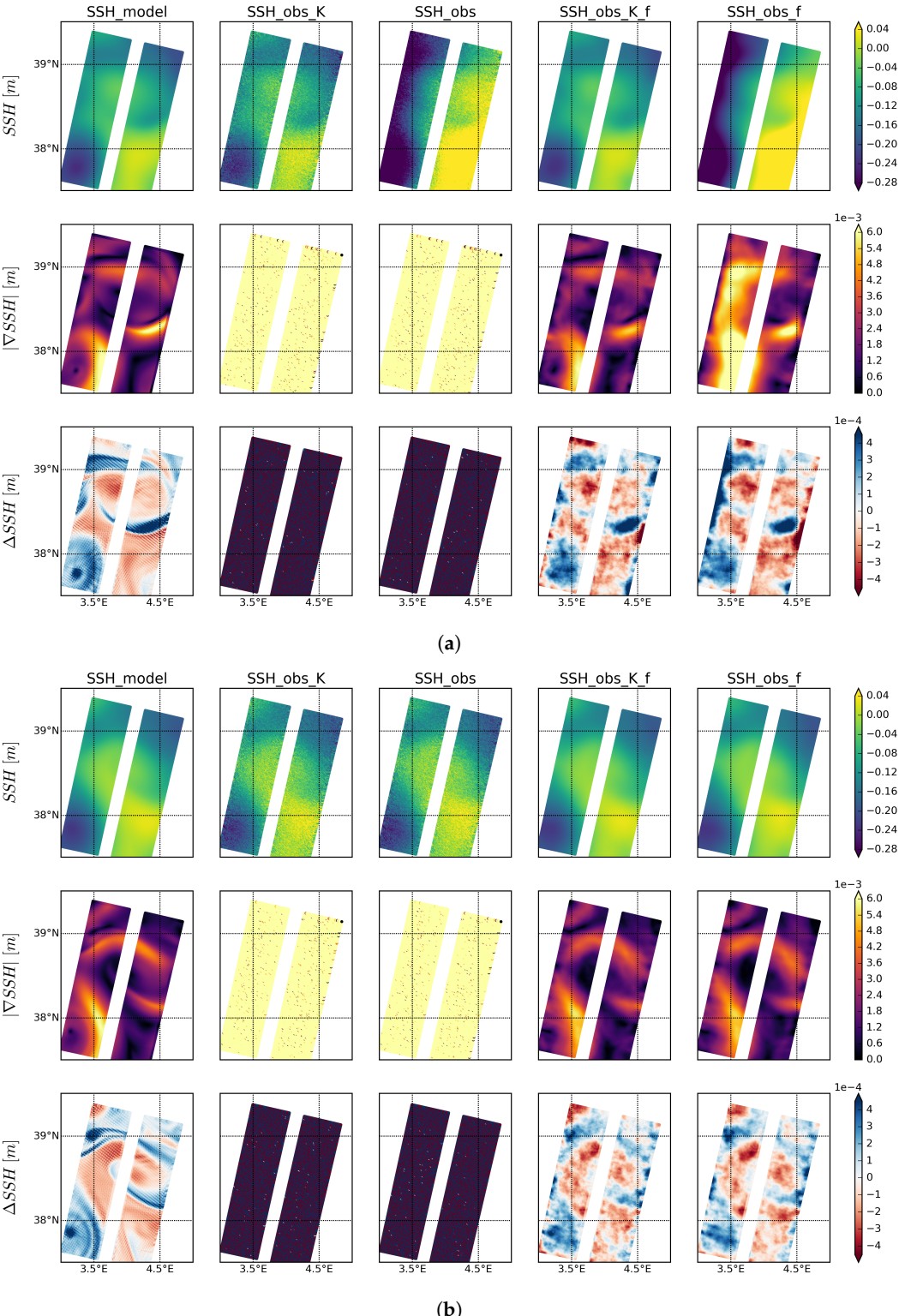

**Figure 5.** Fields of pass 09, cycle 2 (**a**) and 6 (**b**) of the JAS12 dataset compared to the fields filtered with $\lambda_2$ = 355 (455) for SSH_obs_K_f (SSH_obs_f). From top to bottom: SSH, gradient of SSH and Laplacian of SSH. From left to right: model interpolated to SWOT grid (SSH_model), SSH_model + KaRIn noise (SSH_obs_K), SSH_model + all errors (SSH_obs), filtered SSH_obs_K (SSH_obs_K_f), and filtered SSH_obs (SSH_obs_f).

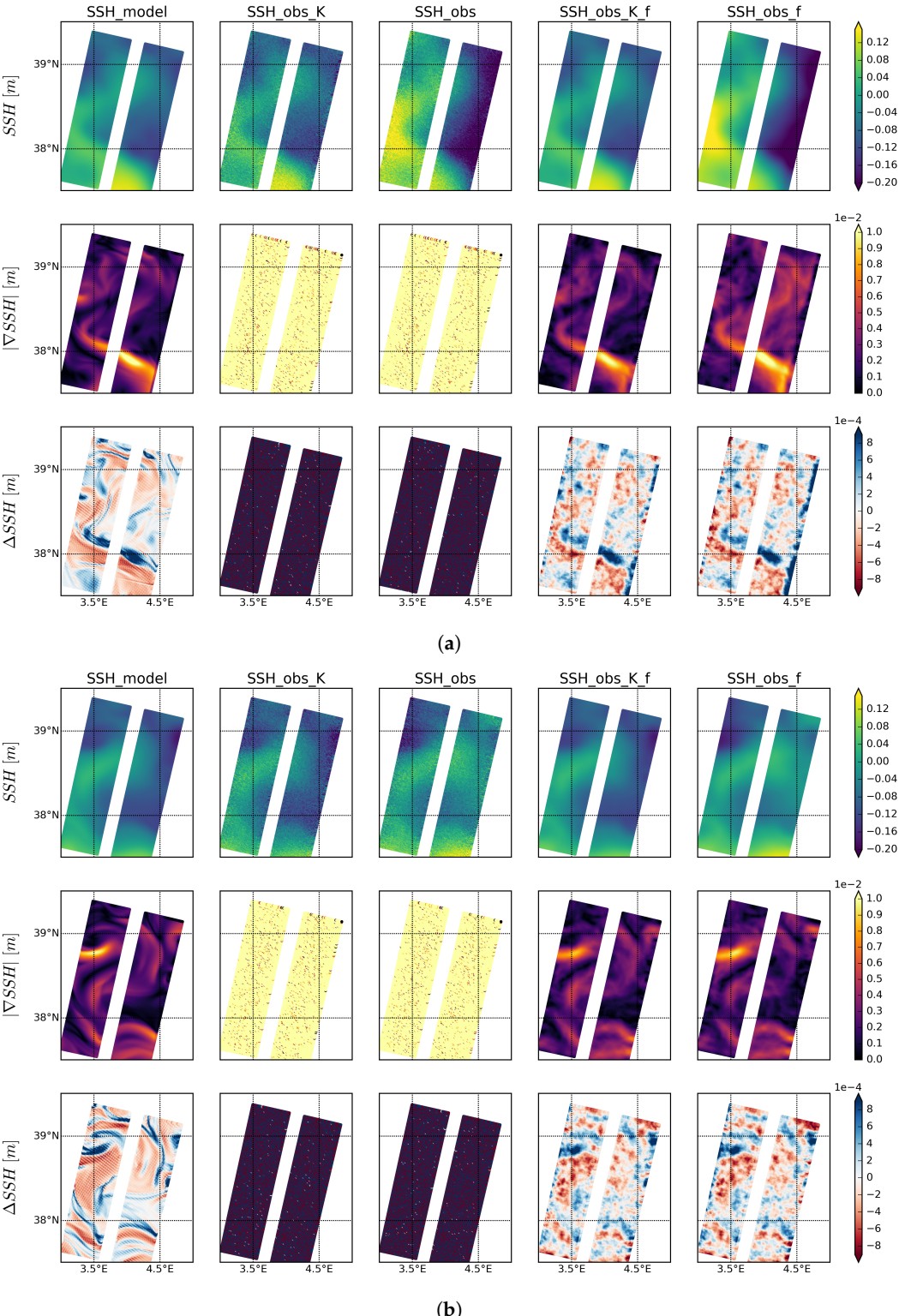

**Figure 6.** Fields of pass 09, cycle 2 (**a**) and 6 (**b**) of the FMA13 dataset compared to the fields filtered both with $\lambda_2 = 105$. From top to bottom: SSH, gradient of SSH and Laplacian of SSH. From left to right: model interpolated to SWOT grid (SSH_model), SSH_model + KaRIn noise (SSH_obs_K), SSH_model + all errors (SSH_obs), filtered SSH_obs_K (SSH_obs_K_f), and filtered SSH_obs (SSH_obs_f).

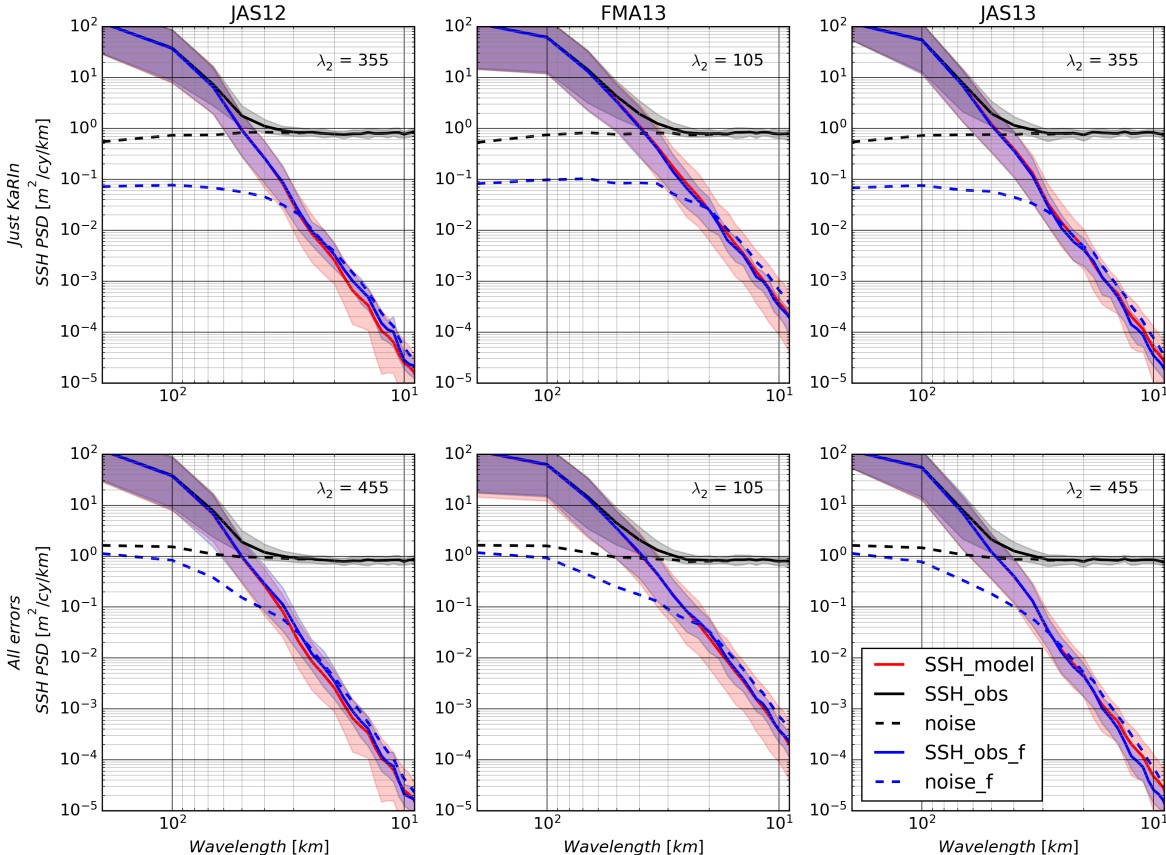

**Figure 7.** Spatial spectra of the model interpolated data (SSH_model) are shown in red, and that of the pseudo-SWOT data (SSH_obs) is in black. Blue lines indicate the filtered pseudo-SWOT spectra (SSH_obs_f) obtained with the same $\lambda_2$ as in Figures 5 and 6. The dashed lines are the noise spectra of SSH_obs (noise) and SSH_obs_f (noise_f). Shaded areas show values between the 5th and 95th percentiles, showing the Power Spectral Density (PSD) variability. Top row shows pseudo-SWOT data with just KaRIn noise added and bottom row shows all errors. Columns represent the different seasonal datasets from left to right: summer 2012 (JAS12), winter 2013 (FMA13), and summer 2013 (JAS13).

## 6. Discussion and Conclusions

Several objectives of the SWOT mission will be met only if the random, small-scale noise and errors affecting the data can be efficiently removed. Small-scale noise, in particular the spatially uncorrelated KaRIn instrument noise, prevents the computation of horizontal SSH derivatives. This limits both the direct estimation of relevant oceanic variables on the SWOT swath and the use of SWOT data to build gridded products of altimetry.

To remove the small-scale SWOT noise, we propose a de-noising method that performs better than conventional convolution-based methods both in terms of RMSE (physical space diagnostic) and spectra. The method, which originates from image processing applications, is based on the regularization of the SWOT SSH data by the penalization of its derivatives of orders 1 to 3 in a variational, optimization framework. This approach is chosen because it is in close connection with the oceanic variables of interest, namely geostrophic velocity and vorticity. After a thorough evaluation based on a large number of simulated SWOT scenes, the variational de-noising method exhibits better performance than standard, boxcar, and Gaussian filters. We find the method performs best when only the second-order derivative ($\lambda_2$) is considered in the cost function. Only one parameter needs to be set, which makes the parameterization of the method as simple as a convolution-based method. We find that this parameter can be set smaller or larger in function of the characteristics of our field: the higher the intensity of the signal, the lower the derivatives penalization needed and thus the value

of the parameter (as we find in the FMA13 $\lambda_2$ values in contrast to JAS12). Also, if the noise level in our fields is higher (all errors scenario), the more we need to penalize and the larger the parameter value. In other words, the higher the signal-to-noise ratio (SNR), the less we need to penalize our field, and so the smaller $\lambda_2$.

The method will require further investigations before operational applications, since we have focused our attention to one particular region (the western Mediterranean Sea), with an ocean circulation free of tidal forcing and a prescribed Significant Wave Height (SWH) of 2 m. The present study shows that, in one single region, the range of optimal parameters changes with the season due to seasonal changes in the ocean surface dynamics. Similar conclusions are certainly expected with respect to regional and dynamical regimes. The NATL60 simulation used here does not include tidal forcing. The behavior and efficiency of the de-noising method may be questioned in the presence of tidal motions and particularly tide-generated internal waves. Finally, the SWH prescribed in the SWOT simulator to compute the KaRIn error amplitude is prescribed to 2 m. As the SWH varies geographically and according to the atmospheric regime, KaRIn errors smaller or larger than those computed for the present study with the SWOT simulator can be expected [36]. The first two aspects (geographic variations of ocean dynamics and internal tides) are presently under study using data from several high-resolution simulations that include tidal forcing: the HYbrid Coordinate Ocean Model (HYCOM) [37], the Massachusetts Institute of Technology general circulation model (MITgcm) [38], and the recent extended NATL60 (eNATL60) simulation (not yet published). This greater range of scenarios will help provide a more generic set of $\lambda_2$s to use in function of the ocean dynamics.

The method should also benefit from additional developments to reconstruct more realistic fields of relative vorticity on the SWOT swath and could ultimately lead to the estimation of vertical velocities. The de-noising process inevitably smooths out the very fine-scale, elongated structures usually visible in surface relative vorticity fields [10]. Restoring these structures should be investigated, for example using appropriate image processing techniques [39,40] or methods already developed in the oceanographic community such as Lagrangian advection [41,42]. Dynamical models could also be used in a data assimilation framework.

To conclude, this de-noising method opens the way to several relevant applications using the SWOT data, possibly including SWOT data validation, assimilation, and SSH mapping. We mention SWOT data validation due to the inpainting capability of the variational de-noising method, i.e., the fact that the process naturally fills the 20-km gap of the SWOT swath (here the gap is inpainted and emptied again after de-noising to restore SWOT data in the original shape). In other words, the SWOT data are interpolated on the track of the SWOT nadir altimeter. This is obviously relevant for data comparison and validation. De-noising is also interesting to preprocess the SWOT data before their assimilation in ocean circulation models. This actually was a primary motivation for the method development. Computing spatial derivatives of the SWOT data allows the implementation of data assimilation methods that account for SWOT error correlations [14,15]. Alternatively, the relative vorticity derived from the de-noising can be directly assimilated. This option has not been explored yet to our knowledge. This de-noising method can also be combined with other techniques to improve the assimilation. We particularly think about combining it with the technique recently developed by Metref et al. [33] to significantly reduce the impact of the geometrically structured, highly correlated SWOT errors (roll, phase, timing, and baseline errors). Finally, this study has been done using the noise and errors simulated by the SWOT simulator version 2.21, and these will very likely be different. We anticipate that the method is simple and flexible enough to be easily adapted to more realistic noise and errors.

**Author Contributions:** Conceptualization, L.G.-N., E.C., J.L.S., and A.P.; formal analysis, L.G.-N.; funding acquisition, E.C., J.L.S., and A.P.; investigation, L.G.-N.; methodology, L.G.-N., E.C., J.L.S., and N.P.; project administration, E.C., J.L.S., and A.P.; software, L.G.-N., E.C., and N.P.; supervision, E.C., J.L.S., and A.P.; validation, E.C., and A.P.; visualization, L.G.-N.; writing—original draft, L.G.-N. and E.C.; writing—review and editing, J.L.S., N.P., and A.P. All authors have read and agreed to the published version of the manuscript.

**Funding:** L.G.-N acknowledges Ph.D. funding from CNES. The contributions of L.G.-N, E.C., J.L.S. and N.P. are also funded by CNES (SWOT Science Team project) and ANR (BOOST-SWOT project, ANR-17-CE01-0009-01). A.P. is supported by the PRE-SWOT project (CTM2016-78607-P).

**Acknowledgments:** L. Gómez-Navarro would like to acknowledge lab colleagues of both institutions for fruitful discussions during this study: Alejandra Rodríguez, Verónica Morales, Evan Mason, Angel Amores, and Eduardo Ramirez at IMEDEA and Sammy Metref, Adekunle Ajayi, Jean-Michel Brankart, and Aurelie Albert at IGE. Also, thanks to IGE intern students that collaborated in this study and that helped to test the code: Nora Poel and Audrey Monsimer. Codes used are available online at the project repository https://github.com/LauraGomezNavarro/paper_Gomez-Navarro_etal_2020.

**Conflicts of Interest:** The authors declare no conflict of interest.

## Abbreviations

The following abbreviations are used in this manuscript:

| | |
|---|---|
| SWOT | Surface Water Ocean Topography |
| SSH | Sea Surface Height |
| FISTA | Fast Iterative Shrinkage-Thresholding Algorithm |
| KaRIn | Ka-band Radar Interferometer |
| RMSE | Root Mean Square Error |
| MSR | Mean Spectral Ratio |
| PSD | Power Spectral Density |
| DP | Derivatives Penalization |

## Appendix A. Calculation of Laplacian

Laplacian are computed using finite differences, following the method proposed by Reference [22]. We note $h$, the image of size $N_x \times N_y$. In a first step, the two components of the gradient are computed as follows ($i = 1, ..., N_x$; $j = 1, ..., N_y$):

$$
\begin{aligned}
(\nabla h)_{i,j}^x &= h_{i+1,j} - h_{i,j} \quad &\text{if} \quad i < N_x \\
&= 0 \quad &\text{if} \quad i = N_x \\
(\nabla h)_{i,j}^y &= h_{i,j+1} - h_{i,j} \quad &\text{if} \quad j < N_y \\
&= 0 \quad &\text{if} \quad j = N_y
\end{aligned}
$$

In a second step, Laplacian is computed as the divergence of the gradient. Divergence of vector $\mathbf{a} = (a^x, a^y)$ is computed as follows:

$$
\text{div}(\mathbf{a}) = b_{i,j}^x + b_{i,j}^y
$$

where:

$$
b_{i,j}^x = \begin{cases}
a_{i,j}^x - a_{i-1,j}^x & \text{if} \quad 1 < i < N_x \\
a_{i,j}^x & \text{if} \quad i = 1 \\
-a_{i-1,j}^x & \text{if} \quad i = N_x
\end{cases}
$$

and

$$
b_{i,j}^y = \begin{cases}
a_{i,j}^y - a_{i,j-1}^y & \text{if} \quad 1 < j < N_y \\
a_{i,j}^y & \text{if} \quad j = 1 \\
-a_{i,j-1}^y & \text{if} \quad j = N_y
\end{cases}
$$

The scheme implemented at the boundaries preserves the image size, contrary to what a standard five-point stencil Laplacian operator would do. Preservation of image size is essential in the gradient descent iterations to end up with a final image of size similar to the initial image.

## Appendix B. FISTA

To speed up the gradient descent iterations, the Fast Iterative Shrinkage-Thresholding Algorithm (FISTA) algorithm [24] is implemented. Setting $t_0 = 1$ and introducing an auxiliary variable $y$ initialized as $y^0 = h^0$, the iterative algorithm of Equation (3) becomes the following :

$$
\begin{aligned}
h^{k+1} &= h^k + \tau \left( m \circ (h_{obs} - y^k) + \lambda_1 \Delta y^k - \lambda_2 \Delta\Delta y^k + \lambda_3 \Delta\Delta\Delta y^k \right) \\
t_{k+1} &= (1 + \sqrt{1 + 4t_k^2})/2 \\
y^{k+1} &= h^{k+1} + \frac{t_k - 1}{t_{k+1}} (h^{k+1} - h^k)
\end{aligned}
\tag{A1}
$$

## Appendix C. Calculation of Spatial Spectra

The spatial spectra used as one of the scores for the de-noising parameterizations are calculated as follows:

1.  Apply a linear detrending;
2.  Remove the spatial mean;
3.  Apply a Tukey window with a 0.5 fraction of the window inside the cosine tapered region;
4.  Compute the 1D spatial Fourier power spectra along-track for each SSH swath across-track dimension.

## Appendix D. Qualitative Figures of Different Methods

To better illustrate the advantage of our de-noising approach, we show in Figures A1 and A2 the fields provided by the boxcar and Gaussian methods, corresponding to the $\lambda_2$ experiments presented in Figures 5 and 6. We only show the all errors scenario. Boxcar derivatives fields are very noisy, as it is specially visible for the Laplacian fields. With the Gaussian method, the Laplacian is less noisy than with our method, but the gradient is over-smoothed.

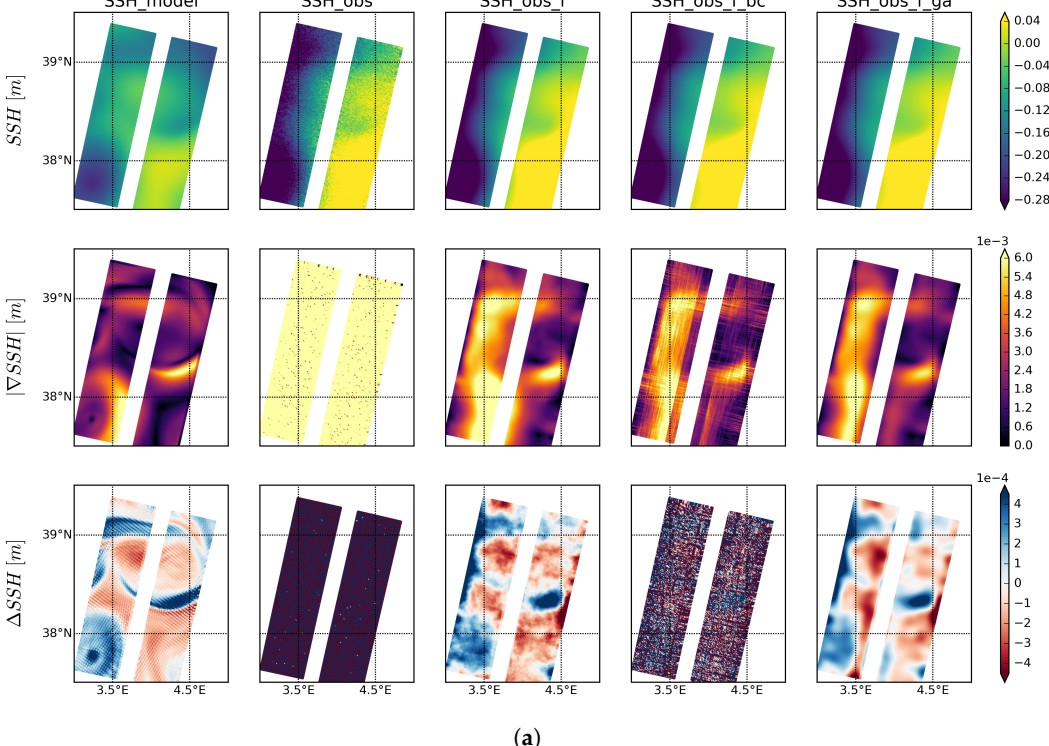

(a)

**Figure A1.** *Cont.*

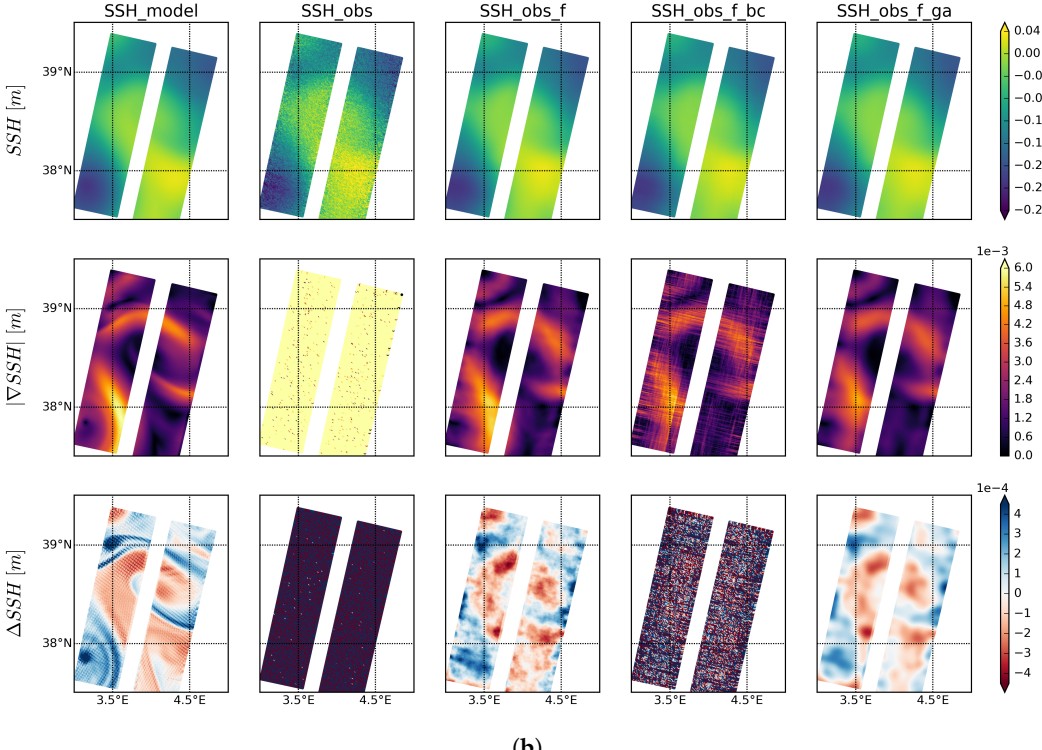

(**b**)

**Figure A1.** Fields of pass 09, cycle 2 (**a**) and 6 (**b**) of the JAS12 all errors dataset. From left to right: comparison between the SSH_model, SSH_obs, SSH_obs filtered with our approach and $\lambda_2 = 455$ (SSH_obs_f), with the optimal boxcar (SSH_obs_f_bc), and with the optimal Gaussian (SSH_obs_f_ga) methods. From top to bottom: SSH, gradient of SSH and Laplacian of SSH

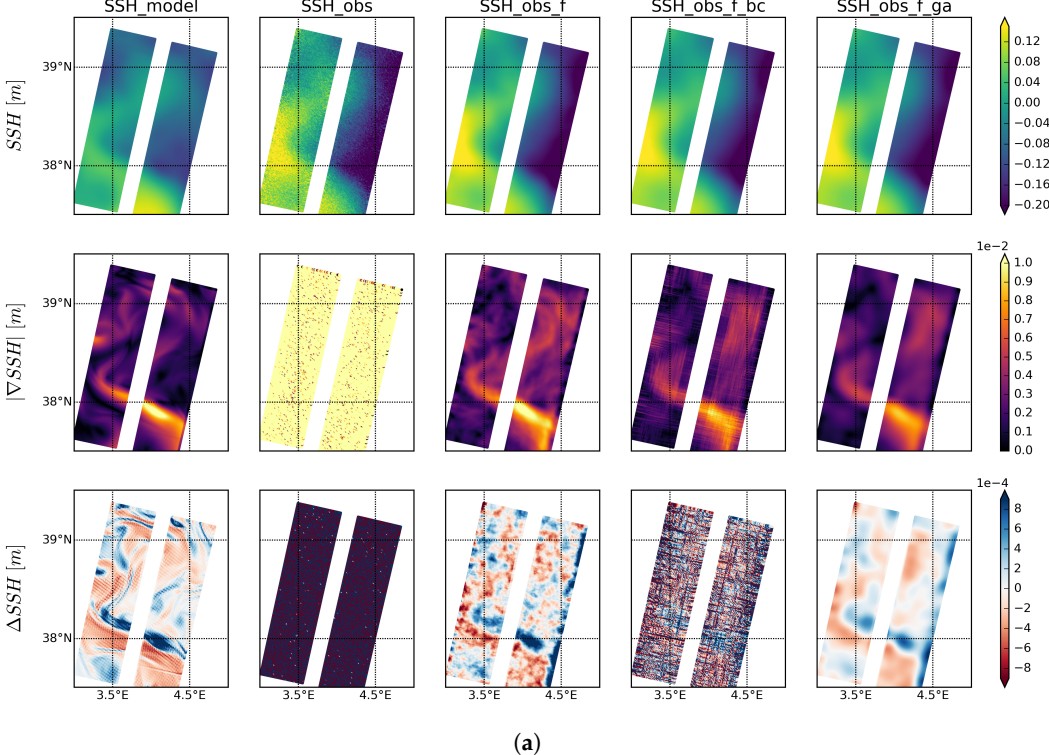

(**a**)

**Figure A2.** *Cont.*

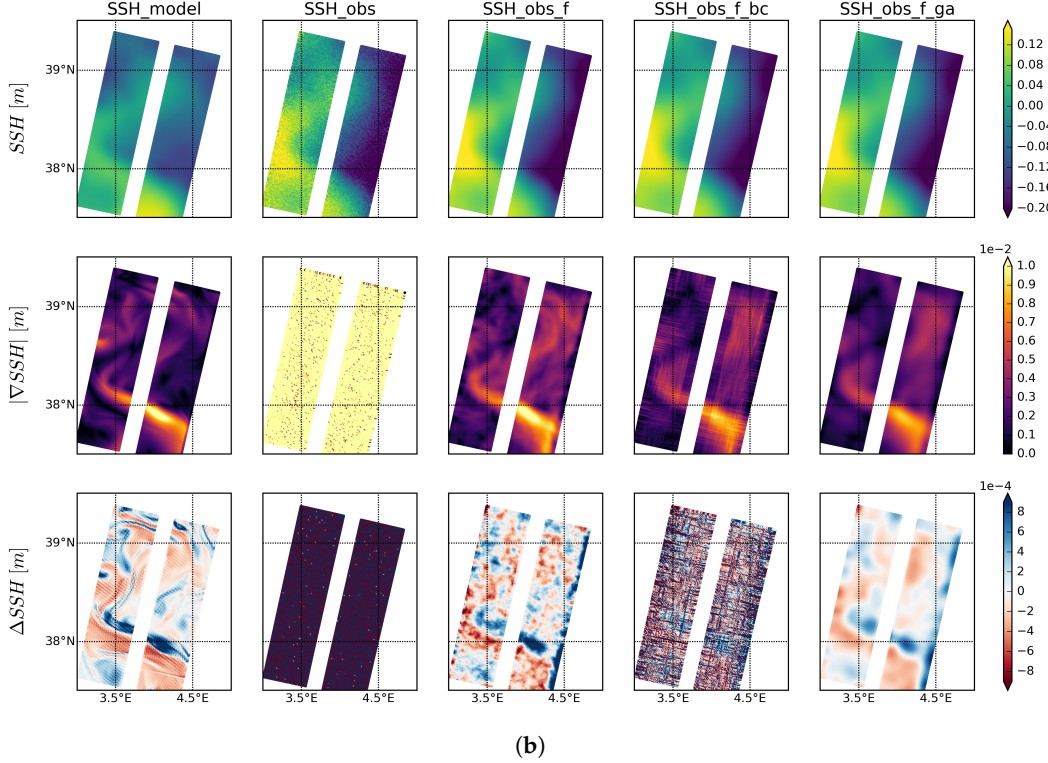

(**b**)

**Figure A2.** Fields of pass 09, cycle 2 (**a**) and 6 (**b**) of the FMA13 all errors dataset. From left to right: comparison between the SSH_model, SSH_obs, SSH_obs filtered with our approach and $\lambda_2$ = 105 (SSH_obs_f), with the optimal boxcar (SSH_obs_f_bc), and with the optimal Gaussian (SSH_obs_f_ga) methods. From top to bottom: SSH, gradient of SSH and Laplacian of SSH

## Appendix E. Softwares

- Standard image techniques: For both boxcar and Gaussian kernel python's scipy.ndimage module was used with the following specific functions:

  - Boxcar filter: *scipy.ndimage.generic_filter*()
  - Gaussian: *scipy.ndimage.gaussian*()

- Variational regularization method: https://github.com/LauraGomezNavarro/SWOTmodule

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
