# Peer review of "Development of an Image De-Noising Method in Preparation for the Surface Water and Ocean Topography Satellite Mission"

_remotesensing, doi:10.3390/rs12040734_

Round 1

Reviewer 1 Report

General comment(s):

I fully understand the importance of filtering the SWOT SSH and pleased to see that this method improves the final results compared to usual filtering methods. However, I have the feeling that something is missing regarding the analysis of your final solution (the one with the lambda2 parametrization). Indeed, as you mentioned and illustrated it, the lambda value depends on the SNR. Thus, this parameter will depend on the characteristics of the area sampled (SSH signal intensity and the sea state which provides a proxy of the noise). It would be thus interesting to determine by how much it is wrong to filter the image with the wrong lambda2 values. If you apply the lambda2 “summer value” to your winter period by how much will you smooth the SSH signal? And on the opposite: how much noise remains (if you filter summer data with winter lambda2 parametrization)?

Then, for users who will want to filter the SWOT SSH, I understand that the problematic will be to define the lambda2 value. Thus, as you mention, the relation between SNR and lambda2 values should be emphasized and further studied to propose a generic law determining the lambda2 values based on the local estimation of the SSH energy (simple to estimate) and the level of random noise (also simple to quantify). I think this an important point to make your method usable.

This filtering problematic is also true for conventional altimetry (it could be mentioned in your paper). I wonder how your method is applicable to conventional altimetry and how such an analysis can bring answer to the limitations induced by simulations (no tides, no waves, …)

I think that the word ‘noise’ used many times for both random noise error and swot spatially correlated errors is confusing. I would recommend the use of the word ‘noise’ for the instrument random error and the naming ‘spatially correlated errors’ to refer to the swot baseline, roll, timing errors. Then it is not clear enough that your objective is not to remove these spatially correlated errors.

To model the SWOT errors you used the SWOT simulator outputs. I think it worth adding one or two sentences to remind that the error simulated are smooth and that the reality might be much more complex. We won’t necessary observed a strong along-track coherence for the spatially correlated errors. You mention that the SWOT errors components are linear and thus reduced when the second derivative is computed: this might be true in the across-track direction but not necessary true for the along-track.

Finally you mention a reduction by factor 2 of the wavelengths observability thanks to the removing of the random noise.  This is an important result but it might be mitigated a bit. Indeed I am not sure that the Wang et al. method can be applied in the case of a signal that have been filtered (I didn’t find the time to propose a demonstration but will think about it).

The conclusion is very good, as mentioned this result is a first step, its open the door to further studies.

 Of course, I will be pleased to discuss all the points mentioned

Reviewer 2 Report

Spatial smoothing provides that the noise has the same spatial resolution as the processed signal.
For satellite altimeria, this is not so.
The moisture correction, calculated according to the microwave radiometer, has a spatial resolution of 25 km, calculated by the model of the atmosphere - 2.5 degrees. This also applies to the “dry” amendment.
The text of the article does not even contain a brief description of how errors in the calculation of amendments are modeled

The article is overloaded with graphic material.
All information is in tables 1 and 2. From figures 5-6 you can choose the most illustrative.

Applications are optional. There are representative things. They can be reduced.

Author Response

-

Reviewer 3 Report

In general, the introduction is well driven. Material-method and the result sections are well described and structured. The discussion and conclusion section is well addressed. Overall, I think that this work could publish in this journal because the authors show a de-noising method that gets better performance than the convolution-based methods.

Author Response

Thank you very much for your time in reviewing our manuscript and for your positive feedback.

Round 2

Reviewer 1 Report

Attached the document corresponding to my second review.

Reviewer 2 Report

The answers to all comments are correct and comprehensive.

Thanks for You

Author Response

Thank you very much.

Round 3

Reviewer 1 Report

I decided to accept the paper as the authors provide answers, elements to all my questions and for all the points that were not clear enough to me.

congratulations to them for this work.